# cATR Tracing Approach to Identify Individual Intermediary Neurons Based on Their Input and Output: A Proof-of-Concept Study Connecting Cerebellum and Central Hubs Implicated in Developmental Disorders

**DOI:** 10.3390/cells11192978

**Published:** 2022-09-24

**Authors:** Willem S. van Hoogstraten, Marit C. C. Lute, Hugo Nusselder, Lieke Kros, Arn M. J. M. van den Maagdenberg, Chris I. De Zeeuw

**Affiliations:** 1Department of Neuroscience, Erasmus Medical Center, 3015 CN Rotterdam, The Netherlands; 2Department of Human Genetics, Leiden University Medical Center, 2333 ZA Leiden, The Netherlands; 3Department of Neurology, Leiden University Medical Center, 2333 ZA Leiden, The Netherlands; 4Netherlands Institute for Neuroscience, NIN-KNAW, 1105 BA Amsterdam, The Netherlands

**Keywords:** disynaptic anatomy, novel anatomical approaches, brain-wide networks, cerebellar developmental disorders

## Abstract

Over the past decades, it has become increasingly clear that many neurodevelopmental disorders can be characterized by aberrations in the neuro-anatomical connectome of intermediary hubs. Yet, despite the advent in unidirectional transsynaptic tracing technologies, we are still lacking an efficient approach to identify individual neurons based on both their precise input and output relations, hampering our ability to elucidate the precise connectome in both the healthy and diseased condition. Here, we bridge this gap by combining anterograde transsynaptic- and retrograde (cATR) tracing in Ai14 reporter mice, using adeno-associated virus serotype 1 expressing Cre and cholera toxin subunit B as the anterograde and retrograde tracer, respectively. We have applied this innovative approach to selectively identify individual neurons in the brainstem that do not only receive input from one or more of the cerebellar nuclei (CN), but also project to the primary motor cortex (M1), the amygdala or the ventral tegmental area (VTA). Cells directly connecting CN to M1 were found mainly in the thalamus, while a large diversity of midbrain and brainstem areas connected the CN to the amygdala or VTA. Our data highlight that cATR allows for specific, yet brain-wide, identification of individual neurons that mediate information from a cerebellar nucleus to the cerebral cortex, amygdala or VTA via a disynaptic pathway. Given that the identified neurons in healthy subjects can be readily quantified, our data also form a solid foundation to make numerical comparisons with mouse mutants suffering from aberrations in their connectome due to a neurodevelopmental disorder.

## 1. Introduction

Disturbances in the development of the cerebellum can lead to a variety of motor and non-motor disorders, such as ataxia and autism [1,2]. Whereas the motor symptoms are mostly characterized by aberrant gate and/or abnormal eye movements [1,3,4], the non-motor symptoms vary from mental dysfunction and cognitive deficiencies to dysregulation of autonomic functioning and affection [5,6,7,8,9]. Like in many other neurodevelopmental disorders [10,11], the symptoms following an abnormal development of the cerebellum probably often reflect, at least partially, aberrations in connectivity [12,13,14]. Indeed, based on MRI studies of cerebellar patients with early symptoms one can expect abnormalities in the output pathways of the cerebellar nuclei with the brainstem [12]. Thus, if one wants to understand the mechanistic underpinnings of such disorders, one needs to know the connectivity of the individual brainstem neurons involved.

However, it has been notoriously difficult to dissect the precise input and output of these intermediary neurons. For example, we know from classical mono-tracer experiments that the cerebellum is connected with neuromodulatory systems such as the ventral tegmental area (VTA), locus coeruleus, and serotonergic centers [15,16]. Yet, we do not know where the individual neurons in these regions that receive input from the cerebellar nuclei in turn project to. Likewise, with more modern viral tracing experiments we and others have been able to uncover that particular thalamic subnuclei project to certain parts of the (pre)frontal cortex to facilitate motor planning [17,18,19], but we can only describe relatively crudely which group of thalamic neurons might be involved, as we do not know for sure which individual neuron is actually relaying the relevant information directly. Unless we are willing to do painstaking whole-cell recordings in vivo or to do structural electron microscopy to show direct monosynaptic contacts [20,21,22,23], we currently have no good way to address this question, let alone to do it efficiently, i.e., at the light microscopic level permitting the identification of individual intermediary neurons in multiple brain regions at the same time. 

So far, light microscopic disynaptic tracing studies have been successful in identifying hubs of clusters of intermediary cells that may connect two different brain regions (Figure 1). For example, one can use cell-type specific tracing of the relationship between input and output neurons (cTRIO) in which Flp-dependent helper virus and Rabies virus Glycoprotein-deleted (RVdG) are injected in the intermediary region and retrograde Cre-dependent virus expressing Flp is injected in the output region of a Cre transgenic animal expressing Cre in a cell-type within the intermediate [24,25,26] (Figure 1A). This approach takes advantage of the fact that the helper virus required for RVdG uptake and transsynaptic spread can be made Flp-dependent. Consequently, RVdG transfection can be limited to those cells expressing Cre and projecting to the specified output area to investigate their brain-wide inputs. Thus, this method is suitable for identifying brain-wide inputs to a specific monosynaptic projection of interest. However, this method does not allow for specific double labeling of the intermediary neurons that connect the input and output areas, and therefore it is difficult to determine unequivocally which of the labeled intermediary neurons is actually responsible for the retrograde labeling in a specific input region. Moreover, as the helper virus has to be injected in a particular intermediary area of interest, one can only screen for one intermediary area at a time, preventing a widespread, let alone whole-brain, screening of such areas in a single experiment.

Similar to RVdG, but concerning anterograde instead of retrograde transsynaptic tracing, is the use of adeno-associated virus serotype 1 (AAV1) [27,28]. When injected in the input location, AAV1 jumps one synapse to all targets and can thereby also label the somata, dendrites and axon terminals of the intermediary neurons as a group (Figure 1B). However, this method does also not allow for the identification of individual intermediary neurons with double labeling; hence if the intermediary neuron is labeled, one cannot be sure that the labeled individual neuron projects to the target area in which transsynaptic fiber labeling occurs, as this labeling might have occurred via neighboring neurons or neurons in other intermediary areas. More specificity can be obtained by adding a Cre-dependent virus in the intermediary region (Figure 1C). For this approach one can combine anterograde transsynaptic tracing of a virus expressing Cre in the input area with Cre-dependent anterograde tracing in the region with the intermediary neurons. This method does allow for the identification of individual intermediary neurons with double labeling, but here too one cannot be sure that the particular individual neuron under the microscope gives rise to the anterograde labeling in the target regions as it may concern an interneuron that did take up Cre, but in fact did not project to the target region. Moreover, this approach is also limited in its efficiency as one has to focus on one intermediate hub at a time. 

Given the limitations of the current methods for disynaptic tracing, we tried to design a new approach that may allow for the unequivocal identification of brain-wide individual intermediary neurons based on both their input and output connections, and that may also allow for a brain-wide approach in which one can identify such neurons in multiple regions at the same time. More specifically, for the current study we combine two existing tracing techniques: anterograde transsynaptic tracing from the cerebellar nuclei (CN) with retrograde tracing from the primary motor cortex (M1), amygdala or VTA in Ai14 reporter mice, using adeno-associated virus serotype 1 expressing Cre (AAV1-Cre) and cholera toxin subunit B (CTB) as the anterograde and retrograde tracer, respectively (Figure 1D). This approach, which we refer to as cATR (combined Anterograde Transsynaptic and Retrograde) tracing, should allow us to double label intermediary neurons in multiple regions from both their input and output side at the light microscopic level. In short, our dataset on the disynaptic projections from the CN to M1, amygdala or VTA show not only that many of the interpretations done by former classical tracing studies can be confirmed [16,29], but also that new pathways can be revealed in a specific and efficient way. Thus, our data show proof of principle of cATR, which allows for a brain-wide dissection of disynaptic connections, in which the intermediary neurons can be identified at the individual cellular level. Moreover, our neuro-anatomical findings on a selection of the disynaptic pathways from the cerebellum provide a detailed quantitative description that can serve as a basis to detect potential aberrations in cerebellar mutants that suffer from neurodevelopmental disorders.

## 2. Materials and Methods

### 2.1. Animals

For the tracing experiments we used homozygous male and female mice of the Ai14 reporter line (No. 007914; Jackson Laboratories, Bar Harbor, ME, USA) [30]. These mice, which were maintained on a C57BL/6 background, express tdTomato fluorescence following Cre-mediated recombination in all cell types, thus including astrocytes. The conditional expression of tdTomato was used to recognize AAV1+ cells. Experiments were performed on mice aged 52–175 days. Mice were group-housed and maintained on a 12 h light–dark cycle with ad libitum food and water. 

### 2.2. Classical and Viral Tracers

CTB (Recombinant; No. C9903 and Alexa Fluor™ 647 Conjugated; No. C34778) were purchased from Sigma-Aldrich (Darmstadt, Germany) and Thermo Fisher Scientific (Invitrogen, Waltham, MA, USA), respectively. CTB and CTB-647 were diluted to 1% in 0.1 M phosphate buffer (PB). Adeno-associated virus AAV1-CMV-HI-eGFP-Cre-WPRE-SV40 (transsynaptic virus expressing Cre and GFP in the nucleus; titer ≥ 8 × 10¹² vg/mL, No. 105545-AAV1) was obtained from Addgene (Watertown, MA, USA). 

### 2.3. Stereotactical Surgery CTB-cATR

To examine projections to M1 and Amy, 120 nL AAV1 was injected in the LCN; (Lambda: −2.3, ML: +2.3, DV: −2.1) of three mice. Two weeks later, 60 nL CTB-647 was injected into contralateral M1 (B: +1.2, ML: −1.75, DV: −0.8) and 60 nL of CTB in the contralateral amygdala (B: −1.2, ML: −2.9, DV: −4.1) of the same mice. For VTA, we performed the same procedure, except that the AAV1 injection was targeted at MCN in three mice (Lambda: −2.7, ML: +1.0, DV: −2.0), and one contralateral injection of 15–60 nL CTB injection was made in the VTA (B: −3.4, ML: −0.35, DV: −4.1) in each of those mice. For two additional mice, the MCN and LCN were injected (coordinates as before) with 80 and 120 nL AAV1, respectively. These injections both spread to interposed nucleus (INT), resulting in primary transfection of all CN (MCN-INT-LCN). Subsequently, we made bilateral VTA injections of 120 nL CTB, each, in those same mice.

For the intracranial injections, mice were deeply anesthetized with isoflurane (1–2%; Isoflutek, Laboratories Karizoo, Barcelona, Spain) and placed in a Stereotaxic apparatus (Kopf Instruments, Tujunga, CA, USA). All mice received subcutaneous injections of 50 µg/kg of buprenorphine (Temgesic, Indivior, Richmond, VA, USA) and 5 mg/kg of Rimadyl (Carprofen, Zoetis, Capelle aan den IJssel, The Netherlands) and their eyes were protected from dehydration using DuraTears (Alcon Laboratories, Geneva, Switzerland). The skull was exposed with a small cutaneous incision and craniotomies were drilled above the CN, M1 and amygdala or CN and VTA, for which the coordinates were obtained from the Paxinos Reference Atlas (Franklin and Paxinos, 1997). Tracer injections were made with pulled glass capillaries (tip outer diameter: ~15 µm; Hirschmann ringcaps, Eberstadt, Germany). Following insertion at 500 µm/minute. the capillary was left in place for 5 min before pressure injection of the tracer at 5–25 nL/minute. To reduce backflow into the pipette tract, the capillary was kept in place for 10 min to allow for diffusion of the tracer in surrounding tissue and was then retracted at 100 µm/minute. After injections, the skull was closed using Histoacryl skin glue. The mice were returned to their home cage individually and monitored for at least 45–60 min while recovering on a heating pad. Mice were transcardially perfused using 4% paraformaldehyde (PFA) 5–7 days after the second surgery.

### 2.4. Immunohistochemistry

Following transcardial perfusion, the brains were dissected and postfixed in 4% PFA for 1.5 h at room temperature (RT). Subsequently, the brains were transferred to 10% sucrose in 0.1 M PB and left overnight at 4 °C, before being embedded in gelatin. The brains were incubated in 12% gelatin/10% sucrose in 0.1 M PB for 30 min at 37 °C, embedded in a plastic mold and left at 4 °C for at least 30 min to harden. They were then cut in small blocks and placed in a 10% formalin/30% sucrose in 0.1 M PB for at least 2.5 h at RT. Embedded brains were transferred to 30% sucrose in 0.1 M PB overnight at 4 °C before being sliced on a Leica SM2000R microtome (Leica Biosystems, Nussloch, Germany) at 50 µm. To prepare the slices for antibody staining, they were blocked for one hour with 10% normal horse serum (NHS) and 0.5% triton in PBS at RT after rinsing in PBS. Subsequently, slices were stained for goat anti-CTB (703; 1:15,000, List Biological Laboratories, Inc., Campbell, NJ, USA) in 2% NHS and 0.4% triton in PBS, and left 24 h at RT. The CTB antibody turned out to be specific enough for unambiguous identification of intensely labeled CTB+ cells on 10× versus background (intrinsic) fluorescence (for which controls have been previously described by our lab [31]). Higher resolution (confocal) imaging did occasionally result in identification of more cells than readily observed under 10× epifluorescence imaging. Next, slices were incubated for 2 h at RT in 2% NHS and 0.4% triton in PBS containing anti-goat-488 secondary antibody (Alexa Fluor, Jackson ImmunoResearch Europe Ltd., Ely, UK) or anti-goat-405 secondary antibody (Alexa Fluor, Jackson ImmunoResearch Europe Ltd., Ely, UK; 1:400 for both). Anti-goat-488 antibody did not interfere with the nuclear localized GFP from the AAV1 virus, since the transsynaptic fluorescence is weak and mainly in the nucleus, in contrast to the granular appearance of CTB in the cytoplasm. The anti-goat-405 antibody resulted in less intense staining and as such was considered less suitable for quantification. When in doubt, cells were not considered CTB+. The slices were rinsed for 40 min before being mounted onto cover slides (VWR) and placed on glass slides with mowiol.

### 2.5. Imaging and Analysis

Initial quantification was performed on images obtained with a fluorescence Zeiss Axio slide scanner with a 10× objective where AAV1 labeled cells presumed to be astrocytes [32], based on their morphology, were excluded from quantification. For amygdala and small VTA-injections, double labeled cells (CTB+/AAV1+) throughout the entire brain were quantified every 1 in 4 slices with a manual meander scan using the grid overlay in FIJI [33] with the cell counter plugin. The projection patterns of LCN, MCN, or MCN-INT-LCN injections helped determine the spread of injection to surrounding nuclei. Since the projections of CN are clearly described in the literature, dense projections functioned as landmarks in our brain-wide analysis. Furthermore, intrinsic fluorescence could often be used (by adjusting brightness/contrast in FIJI until background was visible) for delineation of (sub)nuclei. Where nuclei could not be disentangled consistently, we grouped them together (as is the case for anterior thalamus). Larger VTA injections harbored significantly more transsynaptically labeled neurons, rendering whole-brain quantification unfeasible. Therefore, we visually identified areas with double labeled cells. To illustrate the density of cells in this dataset, a confocal tile-scan and z-stack was made in the superior colliculus (SC) in one of these mice with 40× objective, one airy unit, and optimal z-step according to Nyquist sampling, resulting in 500+ cells quantified in this one scan. Special attention was paid to putative triple labeled (CTB+/CTB-647+/AAV1+) cells. Identification of double labeling at 10× was validated by acquiring confocal z-stacks on a Leica LSM700 with a 40× (1.3 NA) oil objective, 170 nm pixel size in x and y axes, and 3 µm step per slice in the z-axis with 350 µm × 350 µm field of view for amygdala and MCN-VTA. This often demonstrated more cells double labeled in the area than identified at first by 10×. Areas that could not be confirmed with confocal scans were excluded. For M1, because of the reciprocity between thalamus and M1, identification of CTB+ somata could only be achieved with confocal z-stacks. As such, 20× confocal z-stacks and tile-scans at one airy unit, 3 um step size, and optimal × and y resolution were made of 4 slices in each mouse covering VM, CM/CL, and VL, allowing for unilateral quantification of nearly the whole thalamus for the LCN-M1 data. Comparing 20× quantification with 40× quantifications (data not shown) revealed that more double labeled cells can be clearly identified with higher resolution imaging, meaning that our quantifications with 20× confocal z-stacks are likely to be an underestimation. Whole-brain disynaptic projection analysis was done manually by visually identifying fibers in areas that do not traditionally receive CN input and are void of AAV1+ cells. This subjective analysis functions as a method to identify areas with dense disynaptic projections. The simultaneous presence of direct (monosynaptic) fibers and brain-wide AAV1+ cells prevents any automated (objective) analysis.

## 3. Results

### 3.1. Disynaptic Pathways from LCN to M1 and Amygdala

To investigate intermediates that receive input from LCN and project to either M1, amygdala, or both, we used cATR tracing (Figure 1D). AAV1-Cre was injected in the LCN of Ai14 mice, CTB-647 was injected in M1, and CTB in amygdala (for both *n* = 3), (Figure 2A). The LCN injections encompassed the majority of the nucleus itself and, in two of them, reached into the lateral part of the interposed and vestibular nuclei of the cerebellum, and the dorsal part of the cochlear nucleus (Figure 2B,C, middle). The M1 injections were consistently located in the middle along the rostro-caudal axis in layers 5 (L5) and 6 (L6), where one injection ended up more ventrally, focused in L6. All amygdala injections were located in the medial part, focused on CeA and MeA (Figure 2B,C, right). Injection locations were verified based on brain-wide labeling patterns. Primary targeting of LCN was confirmed by the observed presence of AAV1+ cells in its known targets such as red nucleus (RN), mesodiencephalic junction, superior colliculus (SC), zona incerta (ZI), ventromedial (VM), ventrolateral (VL), centromedial (CM) and centrolateral thalamic nucleus (CL) (Appendix A), and by the absence of robust transsynaptic spread to output targets of the neighboring nuclei such as the vestibular (Ve) and parabrachial nuclei (PB). Surprisingly, two specific areas consistently contained labeled cells: the suprachiasmatic nucleus (SCh), which receives input from PB [34], and area postrema (AP), an area without traditional blood–brain barrier (Appendix A). Additionally, in one mouse, we found a small column of cells in S2 in one slice (out of a whole brain, 1 in 2 sections mounted) without the presence of significant anterograde transsynaptic fibers. Considering the absence of fibers and presence of cells until layer 2, a questionable target of LCN, PB, or Ve, this column is unlikely to be the result of transsynaptic spread. Furthermore, AAV1+ cells were occasionally observed in atypical monosynaptic CN targets, such as the cerebral cortex and caudate putamen, possibly due to the previously described risk for AAV1 to cross two synapses [17]. M1 injections resulted in strong retrograde labeling of widespread cortical areas, motor thalamus, CL, parafascicular thalamic nucleus (PF), and posterior thalamic nucleus (Po). Cells found to project to the injection sites in amygdala were mainly located in the ventromedial hypothalamus, paraventricular thalamic nucleus, and anterior insular cortex (Appendix A). As these brain-wide patterns follow previous descriptions in the literature, we continued with our analysis on the intermediates between these regions of interest.

Because of the strong reciprocal connections between M1 and VL, VM, and CL, we focused our analyses on those thalamic nuclei. Detailed quantification of confocal images showed that on average VL seemed to be the preferred pathway for LCN to reach L5 and L6 of M1 (Figure 2D; VL: *n* = 29, CM-CL: *n* = 5, VM: *n* = 12), where the average percentage of double labeled cells over total AAV1+ cells follows a similar pattern (data not shown; VL: 15%; CM-CL: 5%; VM: 8%). Interestingly, the mouse with a more ventral injection of CTB, mostly in L6, showed most double labeled cells in VM, rather than VL (Figure 2D, MCtx2; *n* = 25 and *n* = 10, respectively), which is primarily due to fewer CTB+ (rather than AAV1+) cells being labeled as reflected by the respective percentage of double labeled cells over total AAV1+ cells (data not shown; VM: 11%, VL: 3%). Furthermore, we analyzed the whole brain on 10× for additional intermediate nuclei and observed occasional inconsistent double labeling in PF, Re, and rhomboid nucleus (data not shown). For amygdala, we found consistent but incredibly sparse double labeling in several areas: PAG (*n* = 5 total), deep mesencephalic nucleus (DpMe), (*n* = 7 total), ZI (*n* = 14 total), while cells in CL (*n* = 5 total), and VM (*n* = 7 total) were only found in two mice (Figure 2E). The percentage of double labeled cells over total AAV1+ cells was remarkably low (data not shown; PAG: 2%, DpMe: 1.05%, ZI: 1.92%). Interestingly, specific areas with frequent occurrence of intermingled CTB+ and AAV1+ cells were found, while few double labeled cells (CTB+/AAV1+) were present in LD and lateral hypothalamus (LH) for M1 and amygdala, respectively (Figure 2F). This implies that these areas might receive information from LCN while projecting to M1 and amygdala, but do not serve as a direct intermediate between these two areas. Additionally, triple labeled cells were occasionally found in PF and nucleus reuniens (Re; Figure 2G), indicating that our approach is capable of identifying nodes for divergent information transfer through which the cerebellum provides information via multiple central hubs, in this case to M1 and the amygdala.

### 3.2. Disynaptic Pathways to VTA from MCN versus MCN-INT-LCN

To investigate putative disynaptic projections from MCN to VTA, we next injected AAV1 in MCN and CTb-647 in VTA (Figure 3A) (*n* = 3). The injections in MCN were aimed caudally and dorsally from MCN, to target only the MCN while minimizing leakage to surrounding areas. Following these injections, strong transsynaptic labeling was observed in the PAG, RN, VL, MD/CL, and other thalamic nuclei. Although passage through the RN with the injection pipette cannot be fully prevented, we targeted the VTA injections to the ventral portion to avoid hitting RN as much as possible (Figure 3B,C). Consequently, the injection spot also covered part of the interpeduncular nucleus. We observed strong retrograde labeling in the PB, lateral habenula (LHb) and LH (Appendix A), all major inputs to the VTA [35,36]. Brain-wide quantification showed consistent double labeling (Figure 3E) in several nuclei (Figure 3D): mediodorsal thalamic nucleus (MD; *n* = 31 total), reticulo-tegmental nucleus (RtTg; *n* = 70 total), ZI (*n* = 10 total), LH (*n* = 18 total), PAG (*n* = 11 total), and anterior thalamus (*n* = 37 total). The percentage of double labeled cells with respect to all AAV1+ cells ranged from 9% till 55% (Figure 3D, inset). Surprisingly, dense consistent double labeling was not observed in the ZI or the SC, known targets for CN projections that have been described to strongly project to the VTA [16,24]. However, in additional experiments with larger volume injections covering MCN-INT-LCN (Figure 3F), the subsequent projection patterns include strong double labeling in the same nuclei as the small injections, but additionally show dense double labeling in the ZI and SC (Figure 3G). To assess the strength of these projections when injecting larger volumes, we performed an in-depth quantification of an area with dense AAV1+ and CTB+ cells in SC of one mouse. This showed that of the cells targeted by MCN-INT-LCN (*n* = 351), approximately 10% were double labeled (*n* = 37). This group of double labeled cells made up more than 20% of all quantified SC cells that project to the VTA (*n* = 178), giving insight into the relative strength of cerebellar control over the population of cells in the SC that project to the VTA. Consequently, using our approach, in-depth quantification of large injections can provide a proxy for how strong the influence is from one area over a known projection between two other areas. 

### 3.3. Brain-Wide Disynaptic Cerebellar Projections

Since the axons of the intermediate cells are also labeled by AAV1-Cre (Figure 1B), brain-wide analysis of such transsynaptic fibers can aid confirmation of the presence and estimated strength of the disynaptic pathways through the previously identified intermediates. Additionally, which specific part of the CTB injection spot is targeted by disynaptic cerebellar projections can be readily observed. Moreover, other brain-wide disynaptic targets (through unknown intermediates) can be identified based on observed fibers in areas not receiving monosynaptic input from CN. For VTA, we observed a robust and diffuse projection, not specifically targeting our CTB injection spots (Figure 4Ai,Bi), while a remarkably strong and relatively specific overlap of disynaptic fibers was present within the injection spot in M1 (Figure 4Aii,Bii). Conversely, the fibers in amygdala seemed sparse at best (Figure 4Aiii,Biii). Comparing brain-wide disynaptic projections outside of the injection spots between the LCN, MCN, and MCN-INT-LCN mice illustrated a few interesting patterns. MCN injections showed labeling in mamillary bodies (Figure 4Ci), hypoglossal nucleus (12N; Figure 4Cii), and the bilateral medial facial nucleus (7N), although the 7N projection was stronger on the contralateral side (Figure 4Ciii). Conversely, the LCN had a stronger ipsilateral projection to the whole 7N, with denser labeling in the lateral part (Figure 4Civ). It should be noted that the contralateral MCN to 7N projection could, at least partially, be due to monosynaptic projections from MCN [17,37].

Possibly, cATR also allows for identification of putative trisynaptic pathways without the need for additional tracer injections by investigating areas with strong overlapping retrograde cell labeling and anterograde transsynaptic fibers. In this case, synaptic contact between the strong anterograde disynaptic projection from AAV1 injection spot and retrograde labeled cells (which in turn form the third synapse towards the CTB injection spot) can be inferred by strong overlap or proven with synaptic staining. Notably, we observed dense CTB+ cells in the LHb, a well-known major input to the VTA, without presence of transsynaptically labeled AAV1+ cells. Rather, we observed strong anterograde transsynaptic AAV1+ fibers that co-localized with the dense cell clusters in the LHb, suggesting that the cerebellum targets the VTA not only via a disynaptic pathway, but also via a putative trisynaptic pathway through the LHb, via an unknown intermediate to reach the LHb (Figure 4Di, Figure 5). A similar pattern of intermingled fibers and cells was present in the diagonal band in the basal forebrain (DB), suggesting a second putative trisynaptic projection from cerebellum to VTA (Figure 4Dii, Figure 5). On the other hand, areas of dense retrograde labeling from amygdala (ventromedial hypothalamus and paraventricular thalamic nucleus) did not overlap with strong anterograde disynaptic projections from LCN (Figure 4Diii,Div).

Interestingly, the MCN-INT-LCN injections (CN) did result in anterograde bilateral transsynaptic labeling in LHb (Figure 4Ei), DB (Figure 4Eii), as well as PVT (Figure 4Eiii), while no fibers could be observed in VMH (Figure 4Eiv). Additionally, several disynaptic targets could be observed in these larger injections through unknown intermediates. Specifically, fibers were found in ipsilateral CeA and BMA (Figure 4Fi), and contralateral cingulate cortex (Figure 4ii) and hippocampus (Figure 4Fiii,Fiv).

## 4. Discussion

Here, we utilized a novel tracing approach, cATR, that allows for the identification of known and unknown intermediates at the cellular level through which the CN project to M1 and amygdala, or the VTA. cATR allows us to double label intermediary neurons in multiple brainstem regions from both their input and output side at the light microscopic level. Our dataset on the disynaptic projections from the CN to M1, amygdala or VTA confirms many of the interpretations done by former classical tracing studies and also highlights new disynaptic pathways that can be revealed in a specific and efficient way. Thus, our data show that one can dissect the disynaptic output connections of the CN at a brain-wide level with cATR and at the same time identify the intermediary neurons at the individual cellular level. 

### 4.1. Disynaptic Projections and Trisynaptic Pathways

The different CN showed distinct disynaptic output patterns. Our results suggest that M1-projecting cells that receive input from the LCN were predominantly located in the VL, VM, and CM/CL. For the LCN projection to the amygdala, sparse but consistent intermediate cells could be identified in areas like the DpMe, PAG and ZI, while few anterogradely labeled transsynaptic fibers were visible in the amygdala. Surprisingly, anterograde and retrograde labeling were observed in close proximity without co-localization in the same cell in LD and LH, implying that neighboring separate pathways may exist for M1 and amygdala, respectively. Moreover, our data suggest that in some cases cerebellar output can diverge to two central hubs; for example, cells that receive input from LCN may project to both M1 and amygdala. The disynaptic projections from the MCN to the VTA were mediated by dense cell groups in the anterior thalamus, MD, and RtTg. 

Our findings on the disynaptic projections from the CN to the cortex are largely in line with previous work using classical and viral tracing [18,23,38,39,40]. Yet, in a few cases our data deviated from the classical connections. For example, in one mouse with LCN and M1 injections, the densest labeling of intermediary neurons appeared to be located in the VM, whereas traditionally one expects most intermediary labeling in the VL [40]. This discrepancy may be due to the specific location of the CTB injection for this mouse (MCtx 2), which hit L6 of M1 instead of L5. Since the amount and pattern of AAV+ cells in VL were similar to those in mouse MCtx 3 with many quantified VL cells (Appendix A), the abnormal CTB location appears the most likely explanation. Since all fibers ending up in L2–5 first have to pass through L6, it is possible that the labeling after injection in L6 is not because of CTB uptake by local synapses in L6, but by passing fibers in L6 on their way to L2–5. For example, VM passing fibers may take up CTB more efficiently than VL passing fibers in L6 M1 resulting in stronger labeling in VM. Alternatively, it could be that the data partly represent the preferred pathway for LCN to L6 M1, namely via the VM. Yet, previous work on L5 and L6 M1 input- and output patterns has not highlighted a particularly strong projection from VM to L6, but rather in the opposite direction from L6 cells to VM [41,42]. 

Our findings on the disynaptic projections from the CN to the amygdala and the VTA mimic former studies. Whereas the basolateral amygdala (BLA) has been suggested to receive a disynaptic projection from the CN via the intralaminar nuclei [43], the medial amygdala does indeed not receive a strong disynaptic input from the LCN. Former, anterograde tracing studies from MCN and retrograde tracing from VTA have indicated several potential intermediary targets, including PAG, DpMe, SC, ZI, RtTg, MD, PF, and PB [16,24]. We found MCN connected cells in all these areas, of which the anterior thalamus, MD, and RtTg were most strongly and consistently labeled, presenting themselves as three candidate intermediates through which the CN might modulate VTA activity. The precise nucleus for these cells in the anterior thalamus was hard to determine, as the CL, paracentral, interanterodorsal, and anteromedial thalamic nuclei are located closely together without clearly identifiable borders. Consequently, we grouped them in an anterior thalamic group (Figure 5). Additionally, while we found no primarily labeled AAV1+ cells in LHb, consistent with our expectations, we did observe disynaptically labeled fibers intermingled with dense groups of VTA-projecting CTb+ cells following MCN injections. Consequently, a potentially novel trisynaptic projection from the MCN may also target the VTA. Yet, the precise pathways through which the MCN reaches this potential LHb—VTA projection, remains to be identified. Finally, it should be noted that our side-observations on the disynaptic projections from the cerebellum to hippocampus [44], DB [17], 7N [45,46], and 12N [47], are also in accordance with previous literature. The high level of compatibility of our current data and those obtained previously indicates that we have now a solid foundation to compare data on the output connections of the CN in normal mice with those of mutants that suffer from a cerebellar neurodevelopmental disorder.

### 4.2. Technical Advantages

Our data show proof of principle that cATR allows for a brain-wide dissection of disynaptic connections, in which the intermediary neurons can be identified at the individual cellular level (for advantages and disadvantages compared to other existing multisynaptic approaches, see Table 1). Moreover, our neuro-anatomical findings on the disynaptic pathways from the cerebellum to the cerebral cortex and brainstem provide a detailed quantitative description that can serve as a basis for numerical comparisons with mouse models of cerebellar neurodevelopmental disorders. An interesting technical phenomenon was observed in the LCN to M1 and amygdala data, where most of the CTB+ and transneuronally labeled cells were intermingled, but rarely co-localized within the same cell in LD and LH, respectively. Hence, our approach appears to be able to identify intermingled, non-overlapping neuronal populations. Such quantitative difference might allow us to confirm or reject whether disynaptic connections actually occur in certain areas inferred by monosynaptic tracing. Namely, a common assumption is that if area A projects to B, and B projects to C, that A will project to C via B. However, areas often consist of multiple cell-types with segregated projection patterns [17,48]. Our approach appears to provide an additional tool to further investigate these subareas at a brain-wide level. Additionally, using this approach we also appear able to identify shared intermediaries (Re and PF) that project to two output areas (M1 and amygdala). This showcases that combining diverging (AAV1) and converging (CTB) tracing approaches may reveal the interrelationships between multiple input and output areas as well as their shared and non-shared intermediary centers.

Several additional technical advantages arise from the simplicity of somatic imaging over synaptic identification with high resolution imaging [38,43]. The processing and imaging time required for synaptic identification is many folds higher, rendering brain-wide investigation unfeasible. Besides drastically reducing workload, analyzing disynaptic projections at a cellular level also allows detailed documentation of the precise location of the individual neurons within the intermediate nucleus or subnucleus involved. A similar argument holds for analysis of the injection spot of the output target, which helps to validate the injection site of the retrograde tracer. For example, the disynaptic fibers in M1, coming from all intermediates that were transsynaptically labeled from our AAV1 injection in LCN, overlapped well with the CTB injection spot and mostly stayed clear from the surrounding M2 and S1 regions. On the other hand, disynaptic projections to the VTA appeared more diffuse and there was more spread to the surrounding nuclei, such as the interpeduncular nucleus, RN and SNc, suggesting that the MCN does not target the VTA preferentially with respect to the surrounding areas. 

Last but not least, cATR tracing also allows for objective quantification of the density of labeled intermediary neurons. Although one should always include internal controls to obtain the best estimates, this feature may have advantageous implications for both physiological and clinical studies. For example, with the use of cATR we can now compare the density of the disynaptic projections from the different CN to the VTA with the density of the monosynaptic projections [49], and better assess which inhibitory or excitatory route could be functionally most relevant [50,51,52,53]. Likewise, when using optogenetic stimulation of the CN to interrupt epileptic seizures [54], we can now, based on the numerical data of the disynaptic projections, probably better assess which CN or which intermediary hub is optimally situated for stimulation.

### 4.3. Technical Limitations

Besides these advantages over alternative approaches, several limitations need to be taken into account (Table 1). First, injection sites rarely cover the area of interest completely, without leakage to surrounding areas, which is an inherent problem to all classical and viral tracing. For example, the Re is not a previously described target for LCN [16], while PB has been described to project to Re. Consequently, some PB cells were likely primarily transfected upon injection in CN. Therefore, the few triple labeled cells in Re should not be interpreted as shared intermediates for CN to M1 and amygdala, but should serve as a demonstration that this approach is capable of identifying divergent intermediate nodes on a brain-wide scale. For the dense projections to BMA and CeA in the MCN-INT-LCN mice the external part of the lateral PB (elPB) presents itself as a possible cause. Indeed, our larger MCN-INT-LCN injections leaked more towards PB than the other injections, probably resulting in transfection of elPB, which is strongly (reciprocally) connected to ipsilateral amygdala [55]. The primary transfection of PB may be explained by the white matter anatomy surrounding the CN and PB. The superior cerebellar peduncle comprises the main output of the CN, while it is surrounded by the PB. Consequently, the PB is the preferential direction for the non-Brownian diffusion by which fluid is moving inside the brain. We noted that even when the injection spot was limited to one of the CN, complete absence of primary transfection of PB cells is hard to achieve (data not shown). Likewise, the injection tract reaching the VTA penetrates the RN and other mesodiencephalic areas, posing challenges to the VTA-injections. As such, certain risks inherent to all injections should always be taken into account in the interpretation of anatomical data, such as injection spot leakage, primary transfection of cells outside the area of interest, and the minor capability of AAV1 to jump more than one synapse [17]. Hence, although we aimed all quantified injections relatively dorsal in the CN and ventral for the VTA, our results should be interpreted with caution and only intermediary neurons identified previously with classical anterograde and retrograde tracers should be seriously considered. 

Additionally, the consistent labeling in the SCh and AP might suggest that occasionally some virus ends up in areas not targeted by the injection. The way by which this happens remains to be elucidated, but different possibilities should be considered for these regions. The SCh is a recipient of afferents from the PB, which might have been targeted unintentionally as described above. Likewise, AP is an area without blood–brain barrier, which might have provided a non-neuronal pathway of transport of the virus. Whichever mode of transport is responsible for labeling these neurons, there is no reasonable way by which these cells have a high enough viral load to transsynaptically label their targets [27,28]. Consequently, we do not consider our data on brain-wide intermediates on a cellular level to be affected by these findings.

Another noteworthy limitation is that the projection strength of individual intermediary nuclei to the output can be quantified, but it may be flawed because of the number of collaterals and amount of branching. A smaller number of cells within an intermediary hub could have larger branching (and possibly more synapses) in the target area, which may result in a stronger influence on the postsynaptic cells. As such, the relative relevance of the number of double labeled cells across different nuclei should be interpreted carefully. This net projection strength towards the final output area can be investigated by injecting AAV1-Cre in the starting area of interest, and an anterograde Cre-dependent tracer in the intermediate areas that need to be compared, revealing the branching in the output area (Figure 1C). A similar consideration exists regarding the strength of CN projection towards the intermediates, based on Cre-dependent tracing. However, the current leading hypothesis posits that the electrical ‘strength’ or activity of a projection is correlated to the efficiency of transsynaptic transfection by AAV1 [28]. According to this hypothesis, the number of cells visualized by the current approach likely relates to the amount of branching or electrical activity of the investigated projection. This corresponds to the investigated projections from LCN versus MCN, as they are in accordance with detailed varicosity quantification with classical tracers [16]. Finally, the absolute number of double- or triple labeled cells and anterograde transsynaptic fibers will vary greatly with different injection size and location. Therefore, the focus should lie on the consistently identified areas with significant labeling for each analysis, and if quantification is still a goal, internal control injections should be included in the analysis. 

Given the limitations of CTB labeling, one may facilitate high-throughput brain-wide quantification of the intermediates by replacing CTB with a retrograde AAV, which provides a stronger fluorescent signal [56]. This will also result in less labor-intensive quantification and identification, pushing this approach towards cleared whole-brain imaging and automated analysis [57]. Another reason to develop analogous viral approaches is that different retrograde tracing approaches could result in slightly different results [58]. As such, dedicated investigations on multisynaptic cerebellar projections using various tracing approaches are a necessity in order to verify the true pathways by which regions such as the cerebellar nuclei target brain-wide networks and to relate them to the pathways that have been subject to developmental aberrations. 

## Figures and Tables

**Figure 1 cells-11-02978-f001:**
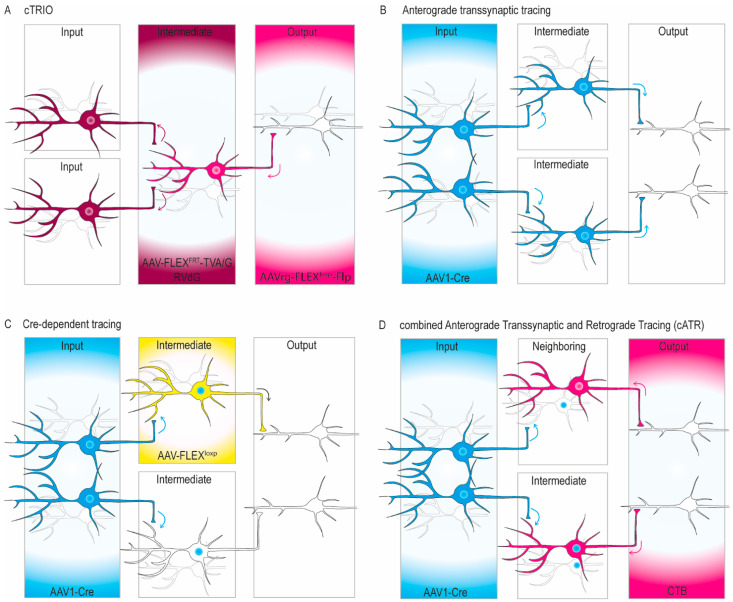
Schematic representations of concepts of currently available disynaptic tracing approaches and cATR. (**A**) Cell-type specific Tracing of the Relationship between Input and Output (cTRIO). In this method a Flp-dependent helper virus (AAV-FLEX^FRT^-TVA/G) and the RVdG are injected in the intermediate and a retrograde Cre-dependent virus expressing Flp (AAVrg-FLEX^loxp^-Flp) in the output area of a Cre-expressing reporter line to facilitate cell-type specificity. Arrows indicate the direction of tracer transport or synaptic jump. This approach allows for the identification of the cells that project to neurons located in the intermediate region targeting the output area, but it does not allow for specific double labeling of the intermediary neurons that connect the input and output areas. Therefore, it is difficult to determine unequivocally which of the labeled intermediary neurons is actually responsible for the retrograde labeling in the input region. Moreover, as the helper virus has to be injected in a particular intermediary of interest, one can only screen for one intermediary at a time, preventing a widespread, let alone whole-brain, screening of such areas. (**B**) Anterograde transsynaptic tracing using an anterograde transsynaptic virus expressing Cre (AAV1-Cre) in the input area in a reporter line, visualizing monosynaptic target cells and their axons; disynaptic projections from injection location, not restricted to one output region. This method does also not allow for the identification of individual intermediary neurons based on a specific output region with double labeling; hence the precise intermediate neuron through which labeling in the output region is observed cannot be unequivocally determined. (**C**) Cre-dependent tracing using an anterograde transsynaptic virus expressing Cre (AAV1-Cre) in the input area, and a Cre-dependent (AAV-FLEX^loxp^) anterograde tracer in a proposed intermediary neuron, similar to (**B**) but more specific. This method does allow for the identification of individual intermediary neurons with double labeling, but here too one cannot be sure that that particular individual neuron gives rise to the anterograde labeling in the target regions as it may concern a local interneuron that did take up Cre, but in fact did not project to the target region. Additionally, alternative intermediates cannot be investigated using this approach. (**D**) Concept of novel approach for disynaptic tracing to identify individual intermediary neurons receiving a specific input and projecting to a particular target region; in this approach the intermediary neurons in between an input and output region are double labeled following a combination of Anterograde Transsynaptic and Retrograde (cATR) tracing. Note that a nucleus could contain both cells receiving information from the input area and cells projecting to the output area, but that these cells are not necessarily the same cell. Instead, they could be neighboring cells (**top**). However, the opposite can also be the case if both tracers end up in the same cell in the intermediate nucleus (**bottom**).

**Figure 2 cells-11-02978-f002:**
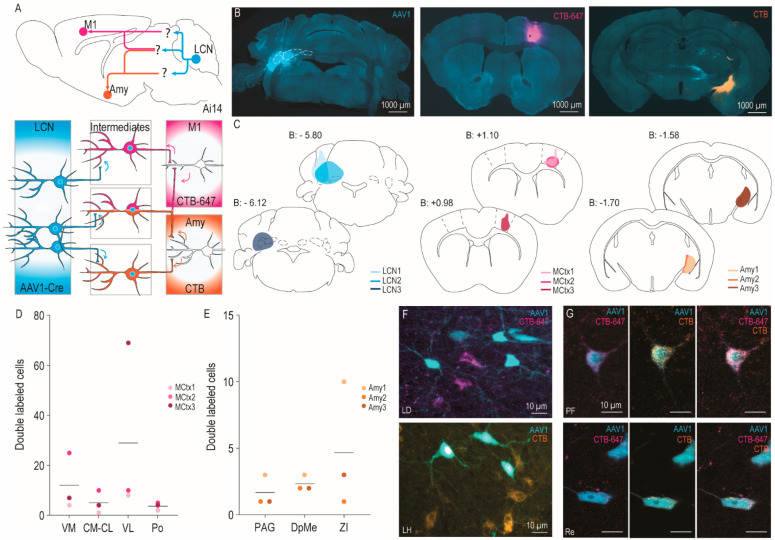
cATR tracing from LCN to M1 and Amygdala. (**A**) Schematic overview of tracing approach demonstrating simultaneous use of three tracers. Blue: AAV1-Cre, Magenta: CTB-647, and Orange: CTB. (**B**) Example picture of injection spots in LCN (**left**), M1 (**middle**), and amygdala (**right**). (**C**) Schematic representation of center of injection spot for LCN, M1, and amygdala for each mouse (*n* = 3), left to right. (**D**) Number of double labeled cells in thalamus between LCN and M1. Horizontal bars reflect the average for all mice, dots are totals for individual mice. (**E**) Number of double labeled cells in three main intermediates between LCN and amygdala. Horizontal bars reflect the average for all mice, dots are totals for individual mice. Note that these quantifications were done on 10× epifluorescence instead of 20× confocal z-stacks. (**F**) Overview of intermingled non-overlapping cell populations for M1 in LD (**top**) and for amygdala in LH (**bottom**). (**G**) Confocal scans of triple labeled cells in PF (**top**) and Re (**bot**). Abbreviations: Amy, amygdala; CM-CL; centromedial and centrolateral thalamic nucleus; DpMe, deep mesencephalic nuclei; CN, cerebellar nuclei; LD, laterodorsal thalamic nucleus; LH, lateral hypothalamus; M1, primary motor cortex; PAG; periaqueductal gray; PF, parafascicular nucleus; Po, posterior thalamic nuclear group; Re, nucleus reuniens; VM, ventromedial thalamic nucleus; VL, ventrolateral thalamic nucleus; ZI, zona incerta.

**Figure 3 cells-11-02978-f003:**
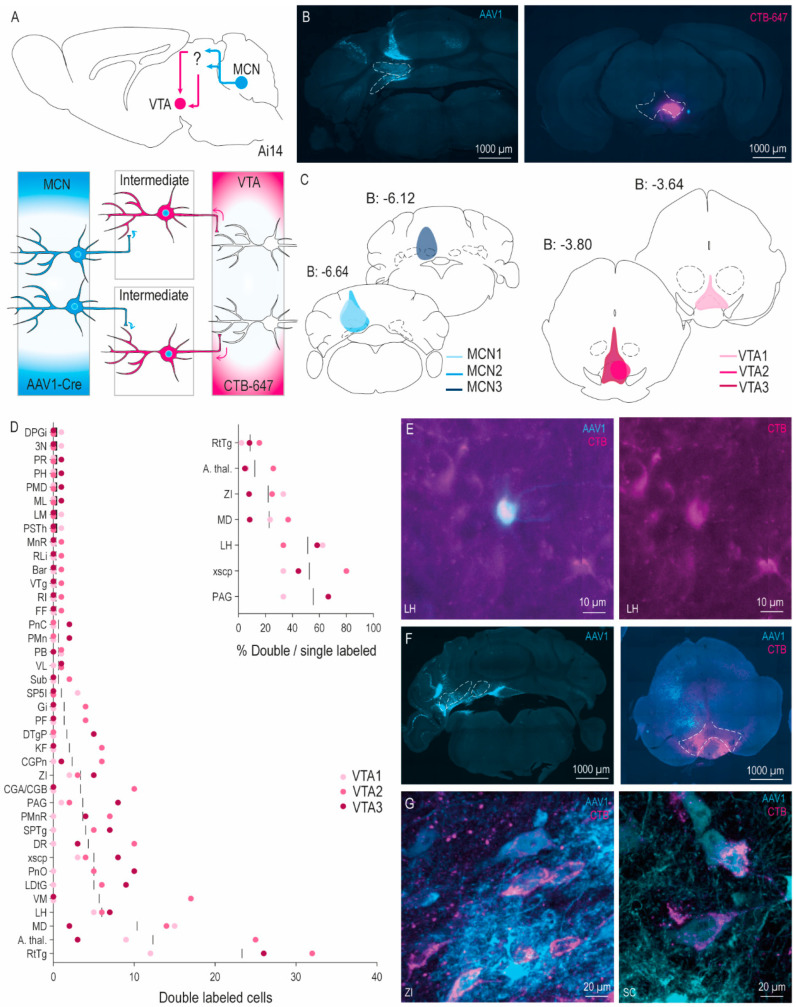
cATR tracing from MCN to VTA. (**A**) Schematic overview of cATR tracing approach from MCN to VTA. Blue: AAV1-Cre, Magenta: CTB-647. (**B**) Example injection spot in MCN (left) and VTA (right). (**C**) Schematic representation of the center of the injection spot of the MCN and VTA for each mouse (*n* = 3). (**D**) Brain-wide analysis of the number of double labeled (AAV1+/CTB+) cells per nucleus per mouse following MCN injections. Vertical bars represent the average double labeled cells per nucleus. Inset shows percentage of double labeled cells over total AAV1+ cells per area in all areas where double labeled cells could consistently be identified. (**E**) Example of double labeled cell in LH. (**F**) Injection spots for MCN-INT-LCN experiments in CN (**left**) and VTA (**right**). Photo of VTA was taken slightly more rostral from center of injection. (**G**) Example photos of double labeled cells in ZI and SC in MCN-INT-LCN experiments. Abbreviations: 3N, oculomotor nucleus; A. thal, anterior thalamic areas; Bar, nucleus of Barrington; CGA/CGB, central gray, alpha and beta part; CGPn, central gray of the pons; CN; MCN-INT-LCN injections; DPGi, dorsal paragigantocellular nucleus; DR, dorsal raphe; DTgP, dorsal tegmental nucleus, pericentral part; FF, nucleus of the fields of Forel; Gi, gigantocellular nucleus; INT; interposed cerebellar nucleus; KF, Kolliker-Fuse nucleus; LC, locus coeruleus; LCN, lateral cerebellar nucleus; LDtG, laterodorsal tegmental nucleus; LH; lateral hypothalamus; LM, lateral mamillary nucleus; MCN, medial cerebellar nucleus; MD, mediodorsal thalamic nucleus; ML, medial mamillary nucleus, lateral part; MnR, median raphe nucleus; PAG, periaqueductal gray; PB, parabrachial nucleus; PF, parafascicular nucleus; PH, posterior hypothalamus; PMD, premamillary nucleus, dorsal part; PMn, paramedian reticular nucleus; PMnR, paramedian raphe nucleus; PnC, caudal pontine nuclei; PnO, pontine reticular nucleus, oral part; PR, prerubral field, PSTh, parasubthalamic nucleus; RLi, rostral linear nucleus; RI, rostral interstitial nucleus of the medial longitudinal fasciculus; SC, superior colliculus; RtTg, reticulotegmental nucleus of the pons; SP5I, spinal trigeminal nucleus, interpolar part; SPTg, subpeduncular tegmental nucleus; Sub, submedius thalamic nucleus; PAG, ventrolateral periaqueductal gray; VL, ventrolateral thalamic nucleus; VM, ventromedial thalamic nucleus; VTA, ventral tegmental area; VTg, ventral tegmental nucleus; xscp, decussation of superior cerebellar peduncle; ZI, zona incerta.

**Figure 4 cells-11-02978-f004:**
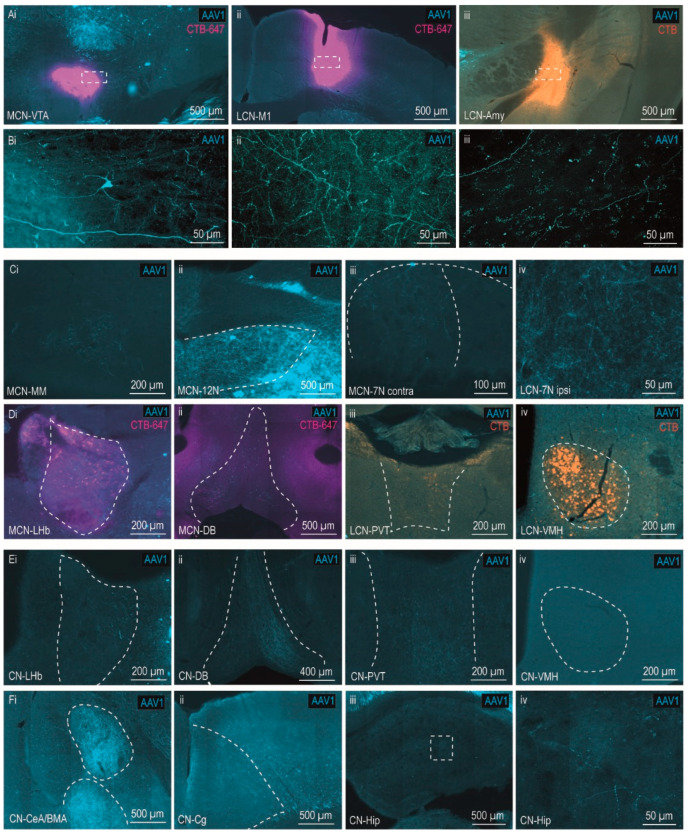
Brain-wide disynaptic projections from MCN, LCN and MCN-INT-LCN (CN). (**Ai**–**iii**) Overview of transsynaptically labeled AAV1+ fibers from the injections shown in Figure 2 and Figure 3, specifically in the CTB injection spots for VTA; (**Ai**), labeling in M1 (**Aii**) and amygdala (**Aiii**). Blue, magenta and orange indicate AAV1-Cre, CTB-647, and CTB, respectively. (**Bi**–**iii**) Confocal scans of transsynaptic labeling (fibers) in areas indicated in A; (**Bi**) 3D image of 20× z-stack in VTA; (**Bii**) 20× confocal image in M1; and (**Biii**) 40× 3D image of 40× z-stack in amygdala. (**Ci**–**iv**) Transsynaptic labeling (AAV1+ fibers) in mammillary bodies (MM) following injection in MCN; (**Ci**), labeling in 12N following injection in MCN; (**Cii**), labeling in contralateral medial 7N following injection in MCN; (**Ciii**), labeling in ipsilateral lateral 7N following injection in LCN (**Civ**). (**Di**–**iv**) putative trisynaptic connections; MCN-VTA cATR tracing shows colocalization of strong anterograde transsynaptic and retrograde labeling in LHb and DB (**Di** and **Dii**, respectively). LCN-amygdala cATR tracing shows absence of anterograde transsynaptic fibers in PVT (**Diii**) and VMH (**Div**). (**Ei**–**iv**) Same areas as in D, but from larger MCN-INT-LCN injections (CN), showing labeling in LHb and DB (**Ei** and **Eii**, respectively) similar to those following MCN injections (**Di** and **Dii**, respectively), strong labeling in PVT (**Eiii**) in contrast to the absence of LCN projections (**Diii**), and absence of fibers in VMH (**Eiv**), similar as following LCN injections (**Div**). (**F**) Disynaptic fibers for MCN-INT-LCN injections (CN) were found in the ipsilateral amygdala (CeA/BMA; **Fi**), contralateral cingulate cortex (Cg; **Fii**), and contralateral hippocampus (Hip; **Fiii**,**Fiv**). Abbreviations: 7N, facial nucleus; 12N, hypoglossal nucleus; Amy, amygdala; BMA; basal medial amygdala; Cg, cingulate cortex; CeA, central nucleus of the amygdala; CN, MCN-INT-LCN injections; DB, diagonal band in basal forebrain, Dien; diencephalon; Hip, hippocampus; LHb, lateral habenula; M1, primary motor cortex; MM, mammillary bodies; PVT, paraventricular thalamic nucleus; VTA, ventral tegmental area.

**Figure 5 cells-11-02978-f005:**
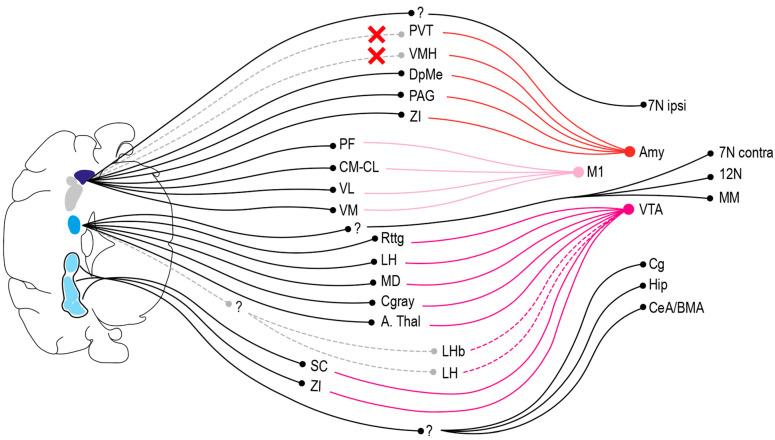
Schematic overview of putative di- and trisynaptic projections from CN to amygdala, M1 and VTA. Graphical summary of the data regarding which specific intermediate neurons relay cerebellar information to M1, amygdala (Amy) and VTA. The results from the MCN and LCN experiments are visualized at the top of the drawing, whereas the results of the MCN-INT-LCN injections are visualized in the bottom. Additionally, unknown intermediates of disynaptic or trisynaptic connections are represented by a ‘?’. Dotted lines represent proposed connections that have yet to be proven, where a red X represents a connection that we failed to identify. This graphical summary is based on the data obtained in these experiments. As such, the absence of a particular connection in this overview is not meant to represent the absence of its existence; only positive results are represented. Abbreviations: 7N, facial nucleus; 12N, hypoglossal nucleus; Amy, amygdala; A. thal, anterior thalamus; Cg, cingulate cortex; Cgray, central gray; CM-CL: centromedial and centrolateral nucleus of the thalamus; DpMe, deep mesencephalic nuclei; Hip, hippocampus; LH, lateral hypothalamus, LHb, lateral habenula; M1, primary motor cortex; MD, medial dorsal thalamic nucleus; MM, mammillary bodies; PAG, periaqueductal gray; PF, parafascicular nucleus, PVT, paraventricular thalamic nucleus; SC, superior colliculus; VM, ventromedial thalamic nucleus; VL, ventrolateral thalamic nucleus; VTA; ventral tegmental area; ZI, zona incerta.

**Table 1 cells-11-02978-t001:** Advantages and disadvantages of cATR compared to other well-known multi-synaptic tracing approaches.

	Suitable Research Question(s)	Advantages	Disadvantages
cATR	What are brain-wide intermediates between area A and B?	Cellular resolution on brain-wide intermediates Short waiting time (2 weeks) Provides stratified information on monosynaptic (AAV1+ cells), disynaptic (AAV1+/CTB+ cells), and trisynaptic (AAV1+ fibers intermingled with CTB+ cells) targets or pathways in one dataset First technique ever to pro-vide information on trisynaptic pathways from A to B with only one unknown stage (see Figure 5) Provides information on other disynaptic targets of A, besides B. This is used to infer trisynaptic pathways (see above and Figure 5)	CTB is also anterograde: fibers can complicate CTB+ cell identification --> confocal imaging was necessary in thalamus Analysis is manual Balance between robustness (sensitivity) and specificity: larger injections lead to larger yield but are less contained to areas of interest Does not provide information on other second-order inputs to B (besides A) No cell-specificity can be achieved in area of interest Transsynaptic labeling with AAV1 occasionally results in labeled cells outside of the known targets for the injected area, possibly because of leakage of the injection spot, disynaptic projections, blood-based transport, or other unknown mechanisms.
cTRIO	What are brain-wide inputs to the direct monosynaptic projection from A to B?	Cellular resolution on brain-wide inputs to cells in A projecting to B Cell-specificity can be obtained with mouse line expressing Cre in specific cell-type in A Analysis is relatively straight-forward since No double labeling needs to be established	RVdG is accompanied by additional safety requirements Not all intermediates between brain-wide inputs and B can be resolved (besides A) Cell-death can occur (RVdG can be toxic)
Traditional rabies	What are brain-wide mono- and multisynaptic inputs to area A?	Cellular resolution on brain-wide mono- and multisynaptic inputs to area A Analysis is relatively straight-forward since no double labeling needs to be established	Specific intermediates between multisynaptic inputs and area A cannot be resolved Cell-death can occur (rabies is toxic) Traditional rabies is accompanied by very strict safety rules
HSV-H126	What are brain-wide mono- and multisynaptic targets of area A?	Cellular resolution on brain-wide mono- and multisynaptic targets of area A	Specific intermediates between area A and multisynaptic targets cannot be resolved HSV is accompanied by strict safety rules

## Data Availability

Data is available upon reasonable request, as total data size exceeds 3TB. Overview files for quantifications performed with FIJI cell-counter plugin and confocal scans under 20GB are available on the following FigShare dois: https://doi.org/10.6084/m9.figshare.20480019.v1 and https://doi.org/10.6084/m9.figshare.20480016.v1.

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
