# Peer review of "cATR Tracing Approach to Identify Individual Intermediary Neurons Based on Their Input and Output: A Proof-of-Concept Study Connecting Cerebellum and Central Hubs Implicated in Developmental Disorders"

_cells, 2022, doi:10.3390/cells11192978_

Round 1

Reviewer 1 Report

In this paper, the authors proposed and tested a new useful combination of neuronal tracing methods (cATR) to identify the relay nucleus for the mutisynaptic projections. For this purpose, the authors used the cerebellar output projections to the motor cortex (M1), amygdala, and ventral tegmental area (VTA). They confirmed many relaying nuclei for these projections and found several new detailed aspects of these projections. They also reported technical limitations of their methods.  This study will give a certain level of contribution to the field of identifying functional neuronal connections in the brain. 

The major concern is that the writing is too concise or not well organized. Consequently, it is difficult to understand the contents systematically.

1)     The contents of the study are not enough consistent; the disynaptic pathway to M1 and amygdala was examined only for the LCN, but not for all CNs, whereas the disynaptic pathway to the VTA was examined virtually only for the MCN. However, the writing (including titles of sections and figures) seems to try to hide this inconsistency as much as possible. This makes the paper very difficult to understand. I would suggest simpler and more straightforward titles and writing. Double labeling data for the amygdala injections should be provided.  

2)     The description of methods is not enough clear in many places.

3)     Panels of figures are not systematically organized.

Methods

Line 128, “Ai14 reporter line”: Specification of Ai14 reporter line should be described for readers. What fluorescent signal does this line express? Are they completely cre-dependent? Does it work in all cells similarly or the efficiency of cre-dependent expression is variable among cells? Which were used, heterozygotes or homozygotes?

Line 136, “AAV1-CMV-HI-eGFP-Cre-WPRE-SV40”

Clearly describe how the labeling was detected for this AAV. Which were detected, the expression of eGFP or the cre-dependent expression of Ai14 reporter?

How the eGFP expression used in the study was not clear. Didn’t it interfere with the labeling of anti-goat-488, which is listed as the secondary antibody.       

Lines 168-171, Specificity of primary antibodies should be described.

Line 177-190, In fluorescent microscopy, the identification of the brain structure in which neurons were counted is not easy since the brain itself is not labeled specifically but has weak intrinsic fluorescence. It should be clearly described how the brain structure was recognized, or how the intrinsic fluorescence signal was captured.  

Line 191, “Whole-brain disynaptic projection analysis was done manually. The meaning is not clear.

Results

The cerebellar cortex areas seem more strongly labeled by the AAV injection (Supplementary Figure 2). Is it possible to quantify the labeling of CN neurons?

The ratio of double-labeled cells and single-labeled cells may be useful information. Please think of providing this in supplemental information.

Section 3.3

In comparison to the title “brain-wide disynaptic cerebellar projections”, the description in this paragraph is quite qualitative. Can the author provide a supplementary table in which the numbers of doulble-labeled neurons in “brain-wide” areas are systematically listed for individual experiments?

Line 273, “(Supplementary Figure 2)”: Does Supplementary Figure 2 have relevant information? Should this be read as “Figure 4”?

Discussion:

Line 397, “VM passing fibers” and “VL passing fibers”: “passing” is very confusing since it means passing through something. Here, I think “VM-originating axons”, “VM-relayed fibers” or something should be better.

Line 440, “the fibers overlapped well with the CTB injection spot…”: Meaning is not clear.

Line 473,

Additionally, the specific labelling in the suprachiasmatic nucleus, AP, and the column in S2 might suggest that occasionally some virus ends up in areas not targeted by the location for injection. This means that in labeling studies with AAV, dense labeling that occurs consistently in multiple injection experiments should be considered meaningful.

Fig. 2

Although the authors wrote “CN” in panel A and the legend, they injected the AAV disynaptic tracer only in the LCN to measure double labeling with the M1 and amygdala. I think “CN” should be replaced with “LCN” in panel A and the legend to avoid confusion.

The graph of double labelled cells between the CN and amygdala is missing.

Fig. 3.

Example pictures from MCN-INT-LCN injection are shown in E. Then, the injection site of this experiment should be shown.

Fig. 4

It is very difficult to understandand this figure. It seems that Ai, Aii, Aiii are from different cases. But, from which case each panel in C-F was obtained is not clear at all. Arrangement of panels should be improved.   

Injection sites in the CN should also be shown.

Fig. 5

Are all identified disynaptic projections mentioned in the text shown here?

The interposed nucleus is drawn in the side opposite to the side where the other cerebellar nuclei are drawn. I do not think it is meaningful. To avoid misleading information, all cerebellar nuclei should be drawn in the same side.  

It is misleading that this graph gives an impression that the projection to the M1/Amygdala starts only from the lateral nucleus, that the projection to the VTA starts only from the medial nucleus and the interpositus nucleus but not from the lateral nucleus, and that the interpositus nucleus provides only ZI and SC projections.  

Red “X” and “?” should be defined in the legend.

Reviewer 2 Report

This study described the application of the “cATR” method to identify neurons that receive inputs from cerebellar nuclei and also project to defined areas of interests, such as the primary motor cortex, the amygdala or the ventral tegmental area. This approach involves the simultaneous injections of two tracers into the Ai14 reporter line, with the AAV1-Cre-mediated anterograde trans-synaptic tracer injected in the cerebellar nuclei, and the retrograde tracer CTB injected into to the primary motor cortex, the amygdala or the ventral tegmental area. The dual labeled neurons represent those intermediate neurons bridging the input area (cerebellar nuclei) and the out areas.

This method allows the identification of the disynaptic pathways originated from cerebellar nuclei. However, it consists of the simultaneous application of two previously-defined tracers, and does not necessarily qualify as a “novel” approach. In addition, the intermediate neurons are identified based on the dual labeling and their anatomical localization, with no indication of labeled cell types. The sensitivity and robustness of identification also relies on imaging with high magnification, which makes it difficult to apply this method at large scale studies.  The authors also attempted to conduct brain-wide analysis of transsynaptic fibers. However, the Ai14 reporter express cell-filling tdtomato after Cre-mediated recombination, and the fibers observed could be from any cells previously infected with Cre.

In summary, this approach helps to reveal intermediate neurons receiving inputs from cerebellar nuclei and also project to defined areas of interest.  More discussion should be placed on the current limitation on the robustness and sensitivity of this approach.

Specific comments:

1.      The authors need to correct their description of cTRIO in Figure 1A. Note that AAV-FLEX-TVA/G represents Cre-dependent AAV helper viruses, whereas cTRIO requires Flp-dependent AAV helper viruses.

2.      The authors mentioned in the introduction the limitation of cTRIO:

“However, this method does not allow for the identification of individual intermediary neurons with double labeling, and therefore one cannot be sure whether a particular intermediary interneuron is the one that leads to the retrograde labeling in the input region.”

The intermediary neurons bridging the input and output area can be precisely identified in cTRIO.  Such neurons must express TVA and G after retrogradely infected with the Cre-dependent Flp viruses, while also expressing the rabies virus-expressed marker. More importantly, cTRIO is meant to be cell-type-specific, and such “starter neurons” in the intermediary areas belong to the same cell type defined by Cre expression.

3.      It would be interesting to list the pros and cons of the cATR method in comparison of polysynaptic tracers, such as wild type rabies viruses and pseudorabies viruses.

4.      Method section should provide enough details about the imaging process. Was the tdTomato native fluorescence used for imaging or additional immunostaining was applied?

Round 2

Reviewer 1 Report

 All my concerns with the original manuscript have been cleared in this revised manuscript. I have only a few minor comments on this manuscript. 

Figure 2 and its legend:

Line 259, “(D) Confocal scans of triple labeled cells in …..”

There seems a mismatch in panels D, E, F, and G between the figure and legend.

Fig 5 legend:

Line 386-, (F) Disynaptic fibers for MCN… injections in amygdala ipsilaterally ……

English seems awkward. It may read “….. in the ipsilateral amygdala……”?

Also in line 369.

Line 515, “some virus ends up in areas not targeted by the injection”. This may be listed in the disadvantage column in Table 1.
